# Multi-Omics Platforms Reveal Synergistic Intestinal Toxicity in Tilapia from Acute Co-Exposure to Polystyrene Microplastics, Sulfamethoxazole, and BDE153

**DOI:** 10.3390/ijms26178441

**Published:** 2025-08-29

**Authors:** Yao Zheng, Jiajia Li, Lihong Li, Gangchun Xu

**Affiliations:** 1Wuxi Fisheries College, Nanjing Agricultural University, Wuxi 214081, China; 13357933502@163.com; 2Key Laboratory of Freshwater Fisheries and Germplasm Resources Utilization, Ministry of Agriculture and Rural Affairs, Freshwater Fisheries Research Center (FFRC), Chinese Academy of Fishery Sciences (CAFS), Wuxi 214081, China; 3College of Fisheries and Life Science, Dalian Ocean University, Dalian 116023, China; lilihong1204@163.com

**Keywords:** intestine, endocytosis, MAPK signaling, actin cytoskeleton regulation, phagosome

## Abstract

Polystyrene microplastic (MP) and its co-existing contaminants may exert different toxic effects on its surrounding aquatic organisms. In order to detect the intestinal harmful responses, tilapia were subjected to exposure with 75 nm of MPs, 100 ng·L^−1^ of sulfamethoxazole (SMZ), 5 ng·L^−1^ of BDE153, and combinations thereof over periods of 2, 4, and 8 days. Enzymatic assays, transcriptomics, proteomics, and metabolomics were employed to evaluate intestinal histopathological effects. Results showed that significant reductions were observed in ATP, ROS, SOD, EROD, lipid metabolism-related enzymes, pro-inflammatory cytokines (TNFα and IL-1β), and apoptosis marker caspase 3 across all groups at day 8. Histological evaluation revealed diminished goblet cell density, with distinct vacuole formation in the BDE153+MPs group. KEGG pathway analysis highlighted disruptions in endocytosis, MAPK signaling, phagosome formation, and actin cytoskeleton regulation. Proteomic findings indicated notable enrichment in endocytosis (decreased sorting nexin-2; increased Si:dkey-13a21.4), MAPK/PPAR signaling, protein processing in the endoplasmic reticulum (Sec61 subunit gamma), and cytoskeletal modulation (reduced fibronectin; elevated activation peptide fragment 1), with or without SMZ and BDE153. Metabolomic profiling showed significant alterations in ABC transporters, aminoacyl-tRNA biosynthesis, protein digestion and absorption, and linoleic acid metabolism. In summary, these findings suggest that BDE153 and MPs synergistically exacerbate intestinal damage and gene/protein expression over time, while SMZ appears to exert an antagonistic, mitigating effect.

## 1. Introduction

Polystyrene is one of the most extensively researched polymer materials, and its microplastic form (MP) is recognized as an effective vector for various environmental pollutants, including antibiotics. These particles are widely dispersed in aquatic environments, with studies reporting average concentrations of 29 ng·L^−1^ in groundwater near drinking-water sources [1]. Antibiotics, in comparison, have been frequently detected in surface waters and sediments at concentrations ranging from 1.12 to 377 ng·L^−1^ and 7.95 to 145 ng·g^−1^, respectively [2]. MPs and antibiotics are widely distributed in aquatic environments, and they can have deleterious effects on a wide range of aquatic species. Given their ubiquitous presence and ecological hazards, the co-occurrence of MPs and antibiotics has become a major environmental concern [1,3]. Several fish experiments as well as studies have shown that organismal damage such as intestinal damage and altered metabolic profiles in fish are associated with the bioaccumulation of MPs and related toxins. The phenomena of their coexistence occur in a lot of study sites based on the collected published data [2]. Sulfamethoxazole (SMZ), a commonly detected antibiotic in aquatic sediments, has been associated with growth inhibition, reproductive disturbances, and physiological abnormalities in aquatic fauna [4]. SMZ has been detected ranging from ng·L^−1^ to μg·L^−1^, and especially in ponds with a higher concentration, such as 273.20 ng·L^−1^ in Taihu Lake [5]. Notably, SMZ in combination with MPs demonstrates heightened toxicity compared to individual exposures, as observed in medaka and zebrafish [5].

MPs are carriers of persistent organic pollutants (POPs) and are capable of absorbing POPs. Several recent studies have assessed the combined effects of MPs and POPs, including antibiotics [1,5]. MPs can accumulate within the gastrointestinal tracts of aquatic animals and may translocate to systemic circulation [6]. POPs pose significant ecological risks due to their chemical stability, tissue-wide distribution, and potential for biomagnification through trophic levels—phenomena reported in the U.S., Canada, and China [7]. MPs also exhibit strong sorptive capacity toward organic pollutants such as polybrominated diphenyl ethers (PBDEs), thereby serving as carriers that can potentiate toxicological outcomes [8]. PBDEs are a class of organic pollutants characterized by low water solubility, high persistence, and widespread distribution [9], with a concentration of 44.0 ng·g^−1^ of lipid in aquatic animals of China. Average concentrations of MP-affiliated ∑8PBDE were 412 ng·g^−1^ in Pearl River Delta, South China, while ∑7PBDEs were from <LOD to 0.78 ng·g^−1^ in the sediment of Yangtze River [10]. The degradation of MPs may facilitate PBDE release into aquatic systems, where they have been found coexisting with crustaceans and beach sediments [9,11].

Environmental data indicate MP concentrations in sediment range from 44.42 to 417.56 items·kg^−1^ [12], while MP-associated PBDEs reach up to 412 ng·g^−1^ in the Pearl River Delta [13]. BDE153 (2,2′,4,4′,5,5′-hexabromodiphenyl ether), a major PBDE congener, has been detected in both human tissues via fish consumption and in aquatic species such as carp [14,15]. Tissue distribution studies show the highest BDE153 concentrations in bile, followed by brain, liver, gills, and muscle [16]. Smaller MPs (~27 μm) are particularly prone to intestinal accumulation [5]. Additionally, microbial biofilms on MP surfaces may enhance pollutant adsorption [17], and studies have identified strong correlations between MPs and total PBDE content. Our previous study showed 10 mg·mL^−1^ 75 nm of MPs resulted in hepatic damage, possibly through PPAR signaling, and an endoplasmic reticulum pathway at 7–14 days [18]. When co-exposed with 5 ng·L^−1^ of BDE153, the enzymatic activities of pro-inflammatory and apoptosis significantly increased, vacuoles appeared, and pathways of endocytosis and the regulation of actin cytoskeleton were significantly enriched [10].

Exposure to MPs and PBDEs can trigger oxidative stress, inflammation, apoptosis, and metabolic disruptions in aquatic organisms, as demonstrated in multi-omics studies [19,20]. BDE153 exposure has also been linked to neurotoxicity [21], and our previous study showed that MPs and BDE153 co-exposure exerted a synergistic toxicity when compared to single exposure [10]. Notably, larger MPs tend to cause damage via indirect inflammatory responses, while nanosized MPs can directly enter cells and induce more severe effects [22]. Despite growing interest, the effects of different exposure durations remain unclear. Extended exposure may affect metabolism, immune responses, and gut microbiota, but these mechanisms are not fully understood [20,23]. Tilapia, a fish species recommended by the FAO, is one of the most widely farmed species globally, with production exceeding 7,000,000 tons, predominantly from China (1,816,828 tons in 2023) and exported worldwide. Therefore, the present study aimed to (1) evaluate the acute intestinal toxicity of MPs, SMZ, and BDE153, both individually and in combination, and (2) elucidate the molecular mechanisms involved using a multi-omics approach including transcriptomics, proteomics, and metabolomics.

## 2. Results

### 2.1. Biological Responses and Enzymatic Activity

At day 2, the levels of ATP (Figure 1a), ROS, and SOD (Figure 1b) showed significant reductions in the SMZ+MP treatment group. By day 8, ATP (also affected at day 4), ROS (with the exception of the SMZ+MP group), SOD, and EROD exhibited marked decreases across all treatment groups (*p* < 0.05). At day 4, significant elevations in total cholesterol (TC) and triglycerides (TG) were observed in the BDE and BDE+MP groups (Figure 1c). Similarly, FAS, LPL, and ACC activities increased significantly in the BDE group at this time point. However, by day 8, all lipid metabolism-related enzyme activities were significantly downregulated in every treatment group (*p* < 0.05). Among them, enzymatic activity levels in the SMZ+MP group remained notably higher than those recorded in the MP, BDE, and BDE+MP groups.

Short-term exposure (2 days) to SMZ and BDE+MP, excluding LPL, led to significant increases in lipid metabolism-associated enzyme activities (*p* < 0.05). No significant changes were found in the MP group relative to the control. Regarding inflammatory and apoptotic markers, TNFα, IL-1β, and caspase 3 levels significantly increased in the BDE+MP group at day 2 (Figure 1d, *p* < 0.05), but these markers declined in all treatment groups by day 8. Interestingly, their concentrations remained significantly higher in the SMZ+MP group compared to other treatments at both day 2 and day 8. Furthermore, TNFα and caspase 3 levels increased significantly in the BDE group at day 4 (Figure 1e, *p* < 0.05).

### 2.2. Transcriptomics

Histological examination revealed a noticeable reduction in goblet cell density across the MP, SMZ+MP, BDE, and BDE+MP treatment groups, with an average decline of approximately 84.0% compared to the control. In the BDE+MP group, cytoplasmic irregular vacuoles of varying sizes were also observed (Figure 2).

Differential gene expression analysis indicated a total of 3269 upregulated and 3575 downregulated genes in the BDE+MP group when comparing day 2 to day 4 (Figure 3a). Substantial transcriptional changes were also recorded in the comparisons of MP versus control (AI8 vs. CI8), BDE versus control (AI8 vs. EI8), and BDE+MP versus control (AI8 vs. FI8) at day 8. Enrichment analysis identified pathways such as focal adhesion, herpes simplex infection, neuroactive ligand–receptor interaction, oxidative phosphorylation, and regulation of the actin cytoskeleton as significantly affected in SMZ+MP versus SMZ comparisons at days 2 and 4. By day 8, and especially in the BDE+MP group, pathways including endocytosis (Figure 3b), herpes simplex virus 1 infection, MAPK signaling, phagosome formation, and actin cytoskeleton regulation were notably enriched.

In the SMZ+MP versus SMZ group, the expression of genes associated with endocytosis (e.g., *igf2r* at day 2, *prkcz* at days 4 and 8, Figure 4 and Figure A1), protein processing in the endoplasmic reticulum (*canx* at day 2), and actin cytoskeleton (*fgfr4* and *fgfr1a3* at day 2) were significantly downregulated via qPCR verification. However, genes such as *mbtps1* and *pdia6*/*sec61g* (except at day 4), and *fgfr1a3* (at days 4 and 8), were upregulated.

For the BDE+MP versus BDE comparison, the expression of *igf2r*, *prkcz*, *canx*, *mbtps1*, *pdia6*, *fgfr4*, and *fgfr1a3* decreased, whereas *sec61g* expression increased at day 2. At day 8, the upregulation of *hgs*, *waslb*, *igf2r* (endocytosis), *canx* and *mbtps1* (endoplasmic reticulum), and *fgfr4* (actin cytoskeleton) was detected in both BDE and BDE+MP groups, while *prkcz, pdia6,* and *myl9a* were downregulated.

Quantitative PCR validation of selected genes related to inflammation and lipid metabolism pathways confirmed the RNA-seq findings and provided further support for pathway-level disruptions.

### 2.3. Proteomics

The proteomics analysis revealed that several biological pathways were significantly enriched in response to treatment, including endocytosis, MAPK/PPAR signaling, protein processing in the endoplasmic reticulum, and regulation of the actin cytoskeleton (Figure 5a, validation in Figure A2).

At day 8, comparisons between the SMZ+MP and SMZ groups showed that key proteins involved in these pathways—such as sorting nexin-2 (Figure 5b, endocytosis), Sec61 subunit gamma (endoplasmic reticulum), and fibronectin (actin cytoskeleton)—were significantly downregulated. In contrast, exposure to BDE+MP compared to BDE alone resulted in the upregulation of proteins, including Si:dkey-13a21.4 (endocytosis, ras-related protein rab7 or its domain-containing protein) and activation peptide fragment 1 (actin cytoskeleton regulation), indicating a notable enhancement effect associated with co-exposure to BDE153 and MPs.

### 2.4. Metabolome

Metabolomic profiling revealed distinct alterations in metabolite abundance across treatment groups (validation in Figure A2). In particular, significant changes were detected in the comparisons between SMZ and SMZ+MP, as well as between BDE and BDE+MP at day 8. The top ten upregulated and downregulated metabolites for each comparison are presented in Figure 6a.

In the SMZ+MP versus SMZ group, several metabolic pathways were significantly enriched, including ABC transporters (Figure 6b), aminoacyl-tRNA biosynthesis, central carbon metabolism in cancer, and protein digestion and absorption. Notable metabolites such as N-acetyl-D-glucosamine, choline, L-cystine, L-proline, L-glutamine, and histamine were markedly reduced (see Figure 6c). For example, decreases in N-acetyl-D-glucosamine, choline, and L-cystine contributed to the enrichment of the ABC transporter pathway, while a reduction in L-proline affected aminoacyl-tRNA biosynthesis, and L-glutamine was linked to disruptions in central carbon metabolism. The downregulation of histamine further contributed to disturbances in protein digestion and absorption.

In the BDE+MP versus BDE comparison, linoleic acid metabolism was significantly enriched. Among the metabolites involved, 12,13-DiHOME and 9(S)-HODE exhibited significant reductions, suggesting that co-exposure with MPs may have intensified BDE153’s impact on lipid-related metabolic pathways.

### 2.5. The Combined Analysis

The correlation analysis of gene expression, protein abundance, and metabolite profiles revealed strong correlations between specific molecular components and key physiological indicators. Several differentially expressed genes (DEGs), including those associated with endocytosis (*hgs*, *waslb*, *igf2r*), protein processing in the endoplasmic reticulum (*canx*, *mbtps1*), and the actin cytoskeleton (*myl9a*), showed statistically significant associations (Figure 7, *p* < 0.05), with enzymatic activities such as ATP, ROS, SOD, EROD, TNFα, IL-1β, and caspase 3. Notably, most DEGs were positively correlated with TNFα and IL-1β, except for *waslb* and *myl9a*.

The metabolite 12,13-DiHOME was strongly associated with EROD levels (*p* < 0.01), suggesting its role in oxidative stress pathways. Similarly, the uncharacterized protein linked to endocytosis showed a significant relationship with EROD expression. In terms of inflammatory and oxidative stress markers, L-proline and 9(S)-HODE both showed highly significant correlations with ATP levels (*p* < 0.01) and moderate associations with IL-1β (*p* < 0.05). These findings suggest that these metabolites may be involved in energy metabolism and inflammatory regulation under co-exposure conditions.

## 3. Discussion

### 3.1. Transcriptomic and Proteomic Pathway Enrichment Highlights Metabolic and Inflammatory Disruptions

The application of multi-omics approaches provided deeper insights into the molecular disruptions induced by microplastic (MP), SMZ, and BDE153 exposure in tilapia. Similarly to previous studies involving crabs and zebrafish, transcriptomic and proteomic profiling identified significant alterations in pathways associated with metabolism and inflammation. In this study, transcriptomic data revealed that pathways such as endocytosis [24], MAPK signaling [25], herpes simplex infection [22], and regulation of the actin cytoskeleton [26] were significantly enriched (Figure 8), in line with earlier findings under MP exposure [10,27]. Within the actin cytoskeleton pathway, upregulation of fibroblast growth factor receptor genes (*fgfr1a3*, *fgfr4*) was detected, echoing reports that exposure to 20 nm of MP (1–100 μg·L^−1^) increases FGF receptor expression in aquatic models [28].

Proteomic results also indicated significant downregulation of fibronectin and protein transport protein Sec61 subunit gamma—findings consistent with those in rat models [29] and previous work from our group [10,18,27,30]. MAPK pathway alterations observed in this study align with earlier reports involving *Daphnia* exposed to MPs [31], confirming its relevance in metabolic stress response mechanisms. MP exposure has been shown to impair intestinal epithelium by inducing oxidative stress, inflammation, and osmoregulatory dysfunction [32,33]. While biofilm-coated MPs were previously reported to increase goblet cell counts in zebrafish intestines [34], other studies observed goblet cell depletion and villus shrinkage [35,36]. Our histological findings supported the latter, with goblet cell loss and vacuolation evident in tilapia intestines after co-exposure.

These structural impairments coincided with increased pro-inflammatory cytokines (TNFα and IL-1β) and apoptotic markers (caspase 3), particularly at earlier exposure stages. Gene expression profiles further validated this inflammatory response, with DEGs related to endocytosis (*hgs*, *waslb*, *igf2r*), endoplasmic reticulum stress (*canx*, *mbtps1*), and actin cytoskeleton regulation (*myl9a*) showing strong correlations with TNFα and IL-1β expression. Moreover, the scale of transcriptional disruption was notably higher at day 8 compared to earlier time points, indicating that prolonged exposure led to more pronounced molecular and histological changes. These outcomes suggest that duration-dependent toxicity plays a key role in MP- and pollutant-induced intestinal impairment.

### 3.2. Lipid Metabolism Alterations Revealed by qPCR and Metabolomic Analysis

Emerging evidence suggests that disruptions to gut microbial metabolism may play a central role in the intestinal toxicity of microplastics (MPs). Previous metabolomic studies in zebrafish exposed to 100 nm of MPs revealed only minor changes under both positive and negative ion detection modes but notable shifts in amino acid and lipid metabolism [33,37]. In this study, significant downregulation of L-proline and L-glutamine was observed following SMZ exposure. L-proline was strongly associated with ATP and IL-1β levels, consistent with prior findings [36], while glutamine downregulation has also been reported in marine diatoms under contaminant stress [38]. Conversely, glutamine upregulation has been noted in the gastrointestinal tissues of zebrafish [39], suggesting species- or tissue-specific responses.

Histamine levels were similarly reduced in both our study and prior research [39]. Since elevated histamine may activate immune responses and contribute to hypersensitivity [40], its reduction may reflect suppressed immune reactivity under pollutant stress. Metabolic pathway analysis revealed significant enrichment of ABC transporters, aminoacyl-tRNA biosynthesis, protein digestion and absorption, and linoleic acid metabolism. Prior studies have linked ABC transporter expression to MP and BDE209 exposure [41,42], and aminoacyl-tRNA biosynthesis has been altered in earthworms and sea cucumbers under similar treatments [43,44]. SMZ exposure has also been shown to affect linoleic acid metabolism, consistent with observations in rice and aquatic organisms [45].

Among the key lipid metabolites, 12,13-DiHOME—secreted by brown adipose tissue—was significantly associated with TNFα expression and is implicated in metabolic disorders such as obesity and dyslipidemia [46]. L-tyrosine levels were increased, while 9(S)-HODE levels were reduced, and choline concentrations were diminished in this study. Since choline is involved in the PPAR signaling pathway [47], these changes suggest the dysregulation of lipid metabolism. In particular, 9(S)-HODE showed a strong association with ATP and IL-1β, indicating a role in energy metabolism and inflammatory modulation. Additionally, the expression of *pparg*, a gene involved in lipid regulation, was significantly reduced under SMZ and BDE153 co-exposure, consistent with recent metabolome-wide association studies [48]. Notably, choline was reported to increase following MP exposure in zebrafish [49], suggesting that co-exposure may disrupt otherwise adaptive metabolic responses. In tilapia, 80 nm of MPs resulted in hepatic lipid metabolism, intestinal microbiota homeostasis, and disorders, when compared to larger sizes of MPs (like 80 µm), and the size was similar to this study (75 nm) [50]. qPCR validation of lipid metabolism-related genes supported these findings, confirming that MPs, BDE153, and SMZ—both individually and in combination—can perturb lipid homeostasis in the tilapia intestine [10], possibly through inflammatory pathways [10,27,50].

### 3.3. Distinct Roles of SMZ and BDE153 in Co-Exposure with Microplastics

Numerous studies have reported intestinal disturbances in aquatic organisms following exposure to MPs, SMZ, and BDE153—either individually or in combination [10,27,36,51,52]. In the present study, combined exposure with SMZ or BDE153 resulted in more severe metabolic and histopathological outcomes than individual treatments. Evidence from enzymatic assays (lipid metabolism and inflammation), transcriptomic and proteomic pathway enrichment, metabolomics, and qPCR validation consistently supported this observation [10,27,50]. Exposure to SMZ and BDE153 led to distinct changes in oxidative stress and lipid metabolism markers. Notably, co-exposure groups showed significant enrichment in pathways related to the regulation of actin cytoskeleton and endocytosis, pointing to their involvement in structural and immune-related intestinal damage. Differences were also apparent in metabolomic profiles. SMZ+MP exposure primarily affected pathways involved in energy production and protein metabolism, while BDE+MP exposure led to pronounced changes in lipid metabolism [10,27,50]. These findings suggest that SMZ may mitigate or modulate certain toxic effects when compared to the synergistic enhancement observed with BDE153.

Previous research has indicated that larger MPs (15 μm) and higher concentrations (500 μg·L^−1^) result in more pronounced intestinal damage in grass carp [52]. In our study, the prolonged exposure to MPs and BDE153—individually or combined—appeared to amplify toxic effects over time, whereas SMZ may have exerted a partial antagonistic influence. This highlights the importance of exposure duration and particle size as critical factors in shaping toxicological outcomes. A limitation of the current study is the lack of mechanistic data explaining how BDE153 intensifies MP-associated toxicity. Understanding this interaction would offer valuable insights into pollution mitigation strategies, particularly in aquaculture settings nearshore or in inland ponds. Growing attention is being paid to remediation technologies, as MPs have been shown to inhibit microbial degradation of polycyclic aromatic hydrocarbons [7] and increase the carcinogenic potential of persistent pollutants [53]. In the field of monitoring and management policies for food health, there should be mandated activities such as bioremediation and the use of carbon-based composite photocatalysts to reduce or eliminate these co-exposure compounds.

Interestingly, plant-derived bioactive compounds such as quercetin and resveratrol have demonstrated protective effects against MP-induced damage [10,18,54,55,56]. Quercetin alleviated liver apoptosis via AMPK/mTOR signaling in grass carp [57], and resveratrol reduced fatty acid synthesis and intestinal injury through the MAPK–PPAR pathway in red tilapia [54,55,56,58]. These findings suggest future potential for dietary interventions or pharmacological mitigation in aquaculture systems.

## 4. Materials and Methods

### 4.1. Sample Collection and Experimental Setup

Polystyrene microplastics (MPs) with a particle size of 75 nm (10 mg·mL^−1^ of stock solution, excitation/emission wavelengths of 488/518 nm, and a concentration of 18,198 × 10^9^ particles·mL^−1^) were purchased and prepared in accordance with procedures as described [18,27,30]. A total of 540 juvenile tilapia (*Oreochromis niloticus*, average weight 26.5 ± 0.6 g) were obtained from the Freshwater Fisheries Research Center (FFRC-CAFS) (Wuxi, China) and randomly allocated into six experimental groups (from group A to group F) in triplicate (each group had three tanks), each tank consisting of 30 individuals. Group A served as the control (CK), Group B was exposed to 100 ng·L^−1^ of sulfamethoxazole (SMZ; C_12_H_14_N_4_O_2_S; dissolved in ddH_2_O, obtained from Shanghai McLean Biochemical Technology Co., Ltd., Shanghai, China), and Group C was exposed to 75 nm of MPs at a concentration of 1.6 × 10^11^ particles·mL^−1^ (spherical particles, negatively charged, with 95.2% falling within the specified size range). Group D received a combined treatment of 100 ng·L^−1^ of SMZ and MPs. Group E was treated with 5 ng·L^−1^ of BDE153 (C_12_H_4_Br_6_O; dissolved in DMSO, with the solvent’s effect considered negligible), and Group F was exposed to both 5 ng·L^−1^ of BDE153 and MPs. Exposure concentrations were selected based on environmentally relevant values reported in the previous literature. SMZ levels were chosen according to measurements in aquaculture water (1.12–377 ng·L^−1^) [59] and comparable studies on zebrafish (5–450 μg·L^−1^) and sea cucumbers (1.2 mg·L^−1^) [43]. The selected BDE153 concentration aligns with prior studies on tilapia exposure [10,27,60].

Throughout the experiment, fish were maintained in aerated water with controlled temperature and quality, using a recirculating culture system supplied by Guangzhou Degang Aquatic Equipment Technology Co., Ltd., Guangzhou, China. Water conditions met the Chinese fishery standards (GB11607-1989), including a pH of 6.9–7.8, dissolved oxygen levels of 5–8 mg·L^−1^, and temperature held at 28 ± 1 °C. All procedures complied with institutional animal ethics guidelines (LAECFFRC-2021-04-08). Sampling was conducted after 2 (*n* = 10), 4 (*n* = 10), and 8 (*n* = 10) days of exposure. Intestinal tissues (*n* = 6 from each tank, the same one) were collected for transcriptomic analysis (*n* = 6, from groups labeled AI8 to FI8), proteomic analysis (*n* = 6, groups A and F), and fecal metabolomic analysis (*n* = 6). Histopathological assessment (*n* = 4 from each tank, the same one), including H&E staining (*n* = 4) and SEM imaging (*n* = 4), as well as biochemical assays (*n* = 4) and qPCR (*n* = 6, the same one as transcriptomic), were carried out using samples from each group.

### 4.2. Histopathological and Biological Determination

Intestinal samples from 12 fish per group (*n* = 4 per tank) were subjected to standard histological procedures, including fixation, embedding, sectioning, staining, and microscopic examination using both light microscopy and scanning electron microscopy (SEM). These procedures followed previously published protocols [10,18,27,30]. For biochemical and enzymatic assays, 0.5 g of intestinal tissue was homogenized, and corresponding supernatants were used for the detection of enzymatic biomarkers. All biochemical analyses (*n* = 4 per tank) were conducted using commercial kits from Nanjing Jiancheng Bioengineering Institute, Nanjing, China, following the manufacturer’s instructions and previous experimental references.

The following biochemical indicators were measured: adenosine triphosphate (ATP, nmol·L^−1^), cytochrome P450 1A1 activity (EROD, pg·mL^−1^, supplied by Jiangsu Meimian Industrial Co., Ltd., Changzhou, China), reactive oxygen species (ROS, IU·L^−1^), superoxide dismutase (SOD, pg·mL^−1^), total triglycerides (TG, μmol·L^−1^), total cholesterol (TC, nmol·L^−1^), fatty acid synthase (FAS, nmol·L^−1^), lipoprotein lipase (LPL, ng·L^−1^), acetyl-CoA carboxylase (ACC, pmol·L^−1^), tumor necrosis factor alpha (TNFα, ng·L^−1^), interleukin-1β (IL-1β, ng·L^−1^), and caspase 3 (pmol·L^−1^). Biochemical parameters were quantified using a spectrophotometer (Jasco-V530, Tokyo, Japan), with absorbance measured within the wavelength range of 340 to 546 nm, depending on the specific reagent and target compound. The experimental design, parameter selection, and procedures were consistent with those reported in earlier studies [10,18,27,30].

### 4.3. Multi-Omics and the Combined Analysis

Intestinal transcriptome, proteome, and fecal metabolome analyses were conducted (*n* = 6) by OE Biotech Co., Ltd. (Shanghai, China) and Nanjing Baokairan Biotechnology Co., Ltd. (Nanjing, China). The experimental workflows and data processing followed methodologies as described [10,18,27,30].

Total RNA was extracted (TRIzol^®^ Reagent, Invitrogen, Carlsbad, CA, USA) and genomic DNA was removed using DNase I (TaKara, Kusatsu, Japan). RNA integrity was evaluated (RNA Nano6000 detection kit, Agilent Bioanalyzer 2100 system, Agilent Technologies, Palo Alto, CA, USA), the RNA concentration was determined (American life technology company, Seattle, WA, USA), and samples with RNA integrity number values larger than 7 were selected. Transcriptome sequencing was performed on intestinal samples collected at 2, 4, and 8 days (*n* = 6 per time point), while proteomic analysis and metabolomic profiling of fecal samples were conducted using samples from the 8-day exposure groups (*n* = 6). We took the transcriptome spliced by Trinity as the reference sequence, and estimated the gene expression level of each sample through RNASeq by Expectation–Maximization (RSEM): aligned clean data to the assembled reference sequence, obtained the read count number of each gene according to the alignment results, standardized the read count data with the trimmed mean of M-values (TMM), and then conducted the analysis with DEGs. With respect to data analysis, raw data removal, gene function annotation, and gene expression estimation, differently expressed genes analysis (DEGs), gene ontology (GO), and Kyoto encyclopedia of genes and genomes (KEGG) enrichment analysis were followed by the reported references [10,18,27,30]. The screening threshold was q-value = 1. Quantitative real-time PCR (qPCR, *n* = 6 per group) was used to validate differentially expressed genes (DEGs) involved in the KEGG pathways related to endocytosis, protein processing in the endoplasmic reticulum, and regulation of the actin cytoskeleton. This study selected *β-actin* as the reference gene, computing changes in mRNA levels. Primer sequences used for qPCR analysis are listed in Table A1. A melt curve analysis was performed at the end of each PCR thermal profile to verify the specificity of each amplicon. The efficiency (E) of each PCR was determined by the slope generated using a 10-fold diluted cDNA series with five dilution points in triplicate. The equation was *E* = 10^(−1/slope)^.

The samples were probe-sonicated in 1 mL of acidified methanol, and the extracts were centrifuged (4 °C, 18,000× *g*, 20 min). For proteomic analysis, supernatants from six intestinal samples per treatment group were subjected to shotgun proteomics using liquid chromatography–mass spectrometry (LC-MS/MS) on a Thermo Fisher Scientific Q-Exactive Focus system in full scan mode (*m*/*z* range 400–1800). Protein concentrations were determined using a bicinchoninic acid (BCA) protein assay kit (Beyotime Biotechnology Co., Shanghai, China) following the manufacturer’s instructions. Mobile phase A was 0.1% formic acid in H_2_O and mobile phase B was 0.1% formic acid in acetonitrile. Chromatographic separation was achieved using a column packed with C18 material (1.7 μm, 2.1 mm × 150 mm) from a Waters UPLC column. The relevant liquid gradient settings were as follows: 0–50 min, B phase linear gradient from 4% to 50%, 50–54 min, B phase linear gradient from 50% to 100%, and 54–60 min, B phase maintained at 100%. Protein identification was carried out using label-free quantification and sequence matching against the UniProt database (downloaded in February 2024, containing 570,157 entries). Proteins significantly associated with the aforementioned pathways were selected for further analysis based on differential expression profiles.

Ethanol (1 mL) was added to the fecal samples (10 μg), which were then subjected to ultrasonic crushing, heated, and then extracted three times using ethanol. The resulting samples were dissolved in 200 μL of solvent (acetonitrile:methanol = 8:2) and centrifuged at 4 °C for 20 min at 10,000 r·min^−1^. The supernatant was then transferred to a C18 solid-phase extraction column. HPLC was performed under the following conditions: a column temperature of 30 °C, a mobile phase flow rate of 0.4 mL·min^−1^, and an injection volume of 5 μL. The mobile phase used formic acid water (pH = 3.25) and β-acetonitrile/methanol (solution B, 8:2) with a mobile phase elution gradient program as follows: 0–1 min, 5% B; 1–3 min, 5–30% B; 3–15 min, 30–100% B; 15–16 min, 100–5% B; 16–17 min, 5% B. Fecal metabolomic profiling was conducted using liquid chromatography–mass spectrometry (LC-MS), as described [18,58]. Secondary metabolites were identified under electrospray ionization mode, using the following parameters: multi-reaction monitoring mode, ion source temperature at 150 °C, desolvation temperature at 550 °C, and desolvation gas flow at 1000 L·h^−1^. Differentially expressed metabolites (DEMs) were filtered using thresholds of fold change (|FC|) > 1, a *p*-value < 0.05, and variable importance in projection (VIP) > 1.

A correlation analysis of DEGs, differentially expressed proteins (DEPs), DEMs, enzymatic activities, and pathway-related biomarkers was performed to explore shared or unique molecular responses, particularly in the 8-day exposure samples. These multi-omics datasets were used to identify key molecular players involved in intestinal toxicity and pathway-specific alterations.

### 4.4. Data Statistical Analysis

All experimental data were processed using SPSS 26.0 software and presented as mean ± standard deviation (SD). Prior to statistical testing, data that did not conform to normal distribution or homogeneity of variance were log_2_-transformed to meet the assumptions of parametric analysis. One-way analysis of variance (ANOVA) was employed to assess differences among treatment groups. When significant variation was detected (*p* < 0.05), Tukey–Kramer post hoc tests were conducted to determine specific group differences. A significance level of *p* < 0.05 was considered statistically meaningful throughout all comparisons.

## 5. Conclusions

Polystyrene microplastics (MPs) act as effective carriers for environmental contaminants such as BDE153 and sulfamethoxazole (SMZ), all of which are frequently detected in aquatic ecosystems. This study investigated the acute intestinal toxicity of MPs alone and in combination with SMZ or BDE153 in tilapia through multi-omics approaches. Significant reductions in ATP, oxidative stress markers, lipid metabolism enzymes, pro-inflammatory cytokines, and apoptosis indicators were observed in all treatment groups after 8 days of exposure. Histopathological assessments revealed a notable decrease in goblet cells and vacuole formation, particularly in the BDE+MP group. Transcriptomic and proteomic analyses showed that pathways related to endocytosis, protein processing in the endoplasmic reticulum, and regulation of the actin cytoskeleton were significantly enriched. Metabolomic profiling indicated disruptions in ABC transporter activity, aminoacyl-tRNA biosynthesis, protein digestion and absorption, and linoleic acid metabolism.

In the comparison between SMZ+MP and SMZ, key proteins (sorting nexin-2, Sec61 subunit gamma, fibronectin) and metabolites (N-acetyl-D-glucosamine, choline, L-cystine, L-proline, L-glutamine, histamine) were significantly downregulated. Conversely, in the BDE+MP versus BDE group, proteins (Si:dkey-13a21.4, activation peptide fragment 1) and metabolites (12,13-DiHOME, 9(S)-HODE) were notably altered, indicating a synergistic toxic effect of BDE153 with MPs over prolonged exposure. In contrast, SMZ appeared to exert a relatively antagonistic or mitigating influence. These findings provide a comprehensive understanding of the distinct and combined toxicological effects of MPs, SMZ, and BDE153, emphasizing the importance of multi-omics tools in evaluating aquatic toxicology. The results offer valuable insights for future environmental risk assessments and potential mitigation strategies in aquaculture and water quality management.

## Figures and Tables

**Figure 1 ijms-26-08441-f001:**
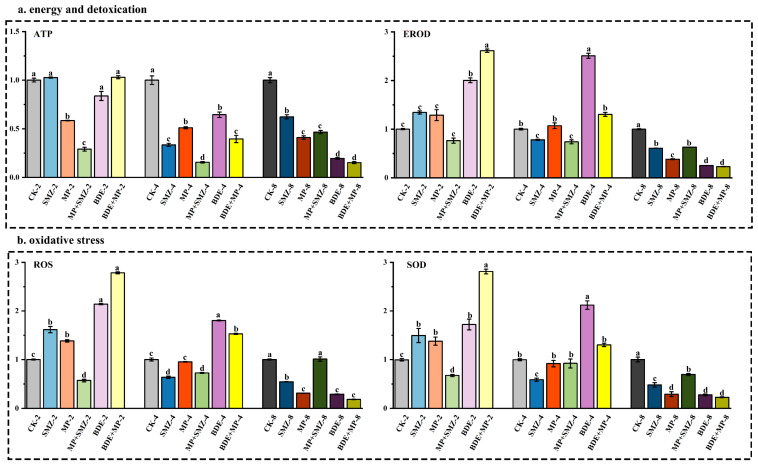
The intestinal enzymatic activities under MP, SMZ, and BDE_153_ co-exposure at 2, 4, and 8 d (*n* = 4). The detected intestinal enzymatic activities have been categorized into five groups, including energy and detoxication, oxidative stress, lipid metabolism, inflammation, and apoptosis. The indexes included TG (μmol·L^−1^), TC (nmol·L^−1^), ROS (IU·L^−1^), ATP (nmol·L^−1^), FAS (nmol·L^−1^), LPL (ng·L^−1^), ACC (pmol·L^−1^), SOD (pg·mL^−1^), EROD (pg·mL^−1^), IL-1β (ng·L^−1^), TNFα (ng·L^−1^), and caspase 3 (pmol·L^−1^). The annotations for each were CK (control), SMZ (100 ng·L^−1^), MP (75 nm, 1.6 × 10^11^ particles·mL^−1^), MP+SMZ (1.6 × 10^11^ particles·mL^−1^ MP plus 100 ng·L^−1^ SMZ co-exposure), BDE (5 ng·L^−1^ BDE153), BDE+MP (1.6 × 10^11^ particles·mL^−1^ MP plus 5 ng·L^−1^ BDE153 co-exposure), and 2/4/8 named as exposure for 2, 4, 8 days, respectively. Different lowercase letters stand for significant level.

**Figure 2 ijms-26-08441-f002:**
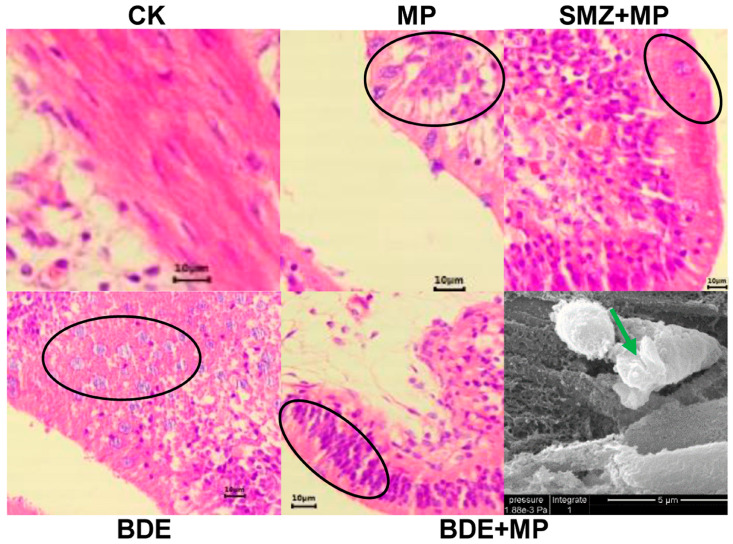
The histopathological changes caused by MP, SMZ, and BDE_153_ co-exposure at 8 d. The black circle shows reduced numbers of goblet cells and the green arrow shows irregular vacuoles in the cytoplasm (*n* = 4).

**Figure 3 ijms-26-08441-f003:**
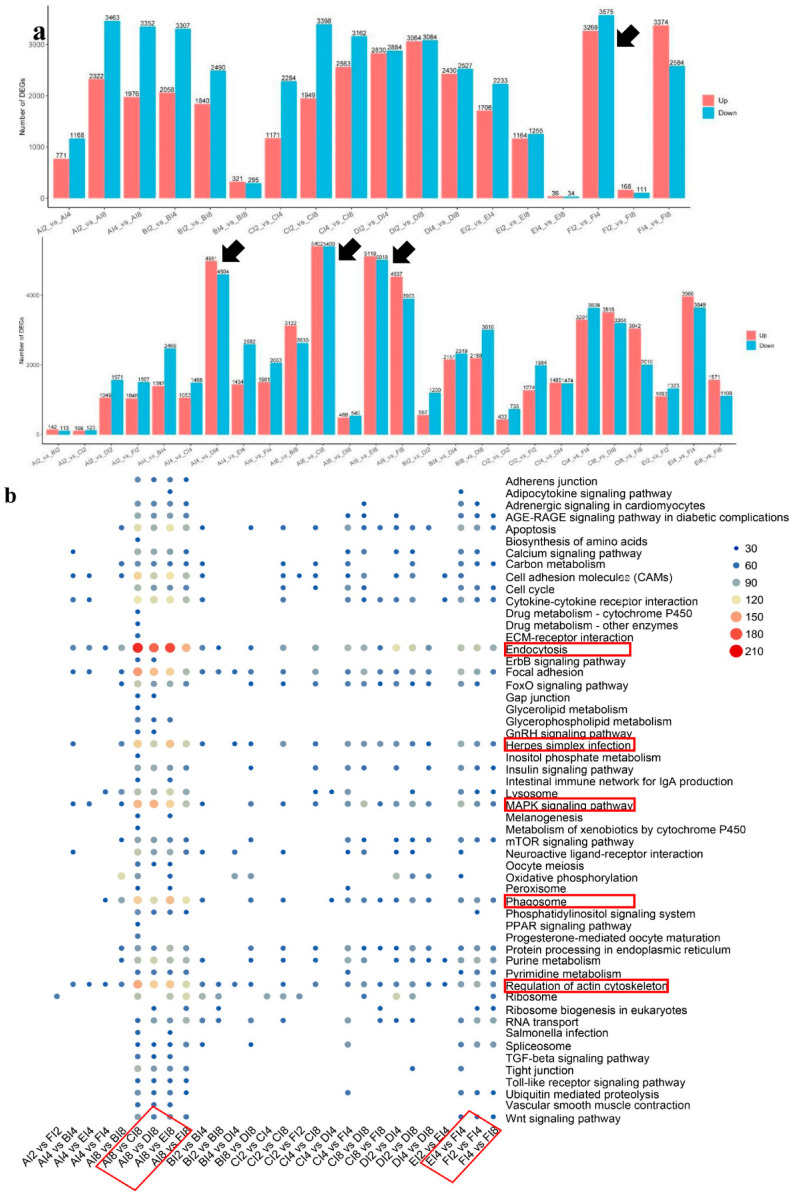
KEGG pathway enrichment caused by MP, SMZ, and BDE_153_ co-exposure at 2, 4, and 8 d (*n* = 6). (**a**), the upregulated (red) and downregulated (blue) DGEs in the comparisons from AI2 to FI8. The annotations for each were CK (control, group A), SMZ (100 ng·L^−1^, group B), MP (75 nm, 1.6 × 10^11^ particles·mL^−1^, group C), MP+SMZ (1.6 × 10^11^ particles·mL^−1^ MP plus 100 ng·L^−1^ SMZ co-exposure, group D), BDE (5 ng·L^−1^ BDE153, group E), BDE+MP (1.6 × 10^11^ particles·mL^−1^ MP plus 5 ng·L^−1^ BDE153 co-exposure, group F), “I” stands for intestine, and 2/4/8 named as exposure for 2, 4, 8 days, respectively; i.e., “AI2_vs. AI4” stands for the comparison in group A between 2 and 4 days. The black arrows showed the larger upregulated and downregulated comparison groups. (**b**) The enriched pathways among the different comparisons. The X and Y axes show different comparisons and different enriched KEGG pathways, and the red boxes show the enriched comparison groups and KEGG pathways. (**c**) The selected DEGs in pathways of endocytosis, herpes simplex virus 1 infection, MAPK signaling pathway, phagosome, and regulation of actin cytoskeleton among different comparisons. The black dot means the enriched pathway in this comparison; i.e., 8-day exposure named as AI8, BI8, CI8, DI8, EI8, and FI8 for 8 days, respectively, intestine for transcriptome; from AI8 to FI8 for group A and F, intestine for proteome and fecal contents for metabolome; “I” stands for intestine. In the Y axis, the length of the black rectangles stands for the values for the enriched DEGs, while the black spot means the hit for this enriched KEGG pathway.

**Figure 4 ijms-26-08441-f004:**
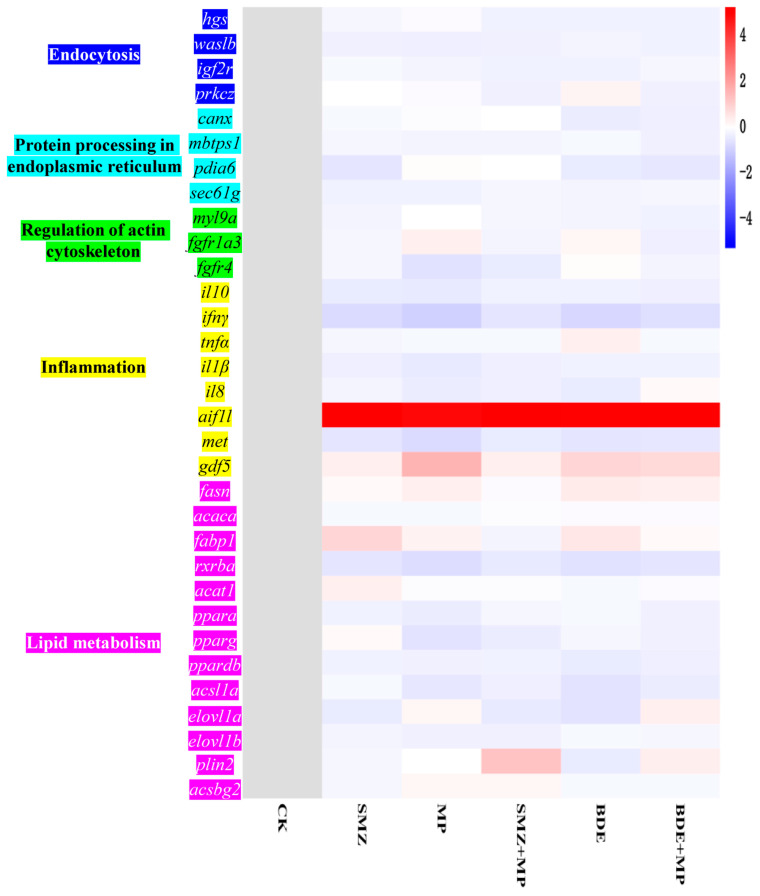
Gene verification by qPCR (*n* = 6, *p* < 0.05 stands for the significance level). Endocytosis: *hgs*: hepatocyte growth factor-regulated tyrosine kinase substrate isoform X1, *waslb*: neural Wiskott–Aldrich syndrome protein, *igf2r*: cation-independent mannose-6-phosphate receptor, and *prkcz*: protein kinase C zeta type isoform X1. Protein processing in endoplasmic reticulum: *canx*: calnexin, *mbtps1*: membrane-bound transcription factor site-1 protease, *pdia6*: protein disulfide-isomerase A6, and *sec61g*: protein transport protein Sec61 subunit gamma. Regulation of actin cytoskeleton: *myl9a*: myosin regulatory light polypeptide 9, *fgfr1a3*: fibroblast growth factor receptor 1-A isoform X3, and *fgfr4*: fibroblast growth factor receptor 4. Inflammation: *il10*: interleukin 10, *ifnγ*: Interferon γ, *tnfα*: tumor necrosis factor α, *il1β*: interleukin 1β, *il8*: interleukin 8, *aif1l*: allograft inflammatory factor 1-like, *met*: MET proto-oncogene, receptor tyrosine kinase, and *gdf5*: growth differentiation factor 5. Lipid metabolism: *fasn*: fatty acid synthase, *acaca*: Acetyl-CoA carboxylase, *fabp1*: fatty acid-binding protein 1, *rxrba*: retinoic acid receptor RXR-beta-A, *acat1*: acetyl-CoA acetyltransferase 1, *ppara*: peroxisome proliferator-activated receptor alpha, *pparg*: peroxisome proliferator-activated receptor gamma, *ppardb*: peroxisome proliferator-activated receptor delta, *acsl1a*: acyl-CoA synthetase long-chain family member 1a, *elovl1a*: ELOVL fatty acid elongase 1a, *elovl1b*: ELOVL fatty acid elongase 1b, *plin2*: perilipin 2, and *acsbg2*: acyl-CoA synthetase bubblegum family member 2.

**Figure 5 ijms-26-08441-f005:**
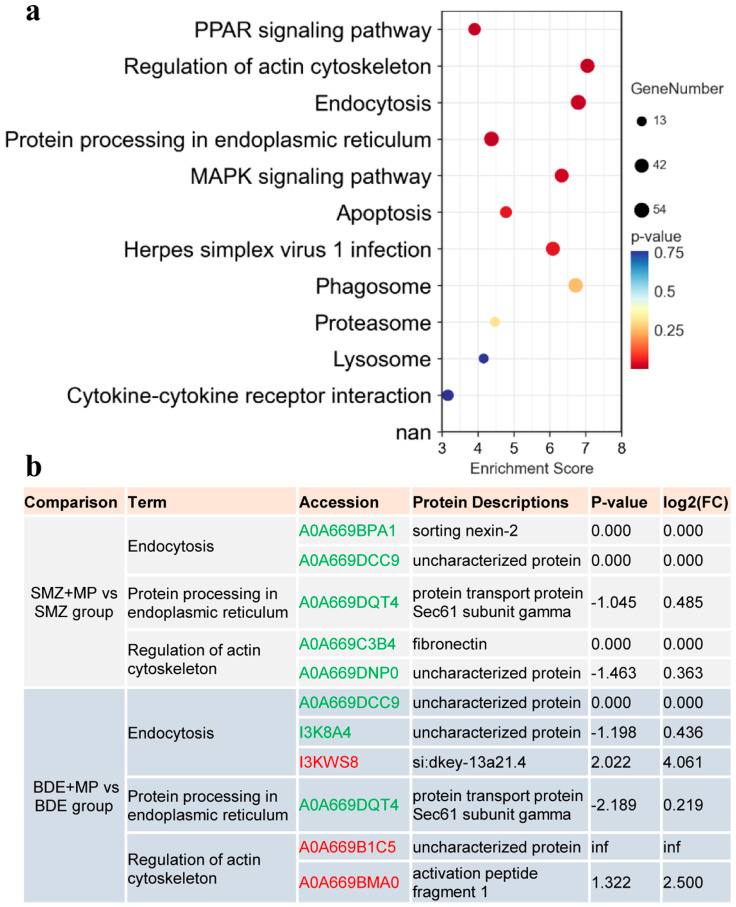
The proteomics result caused by MPs and co-exposure with SMZ, BDE_153_ at 8 d (*n* = 6). (**a**) KEGG pathway enrichment among the comparisons. X and Y axes showed different enrichment scores and different enriched KEGG pathways after selection based on the values for lists and adjusted *p* values. (**b**) The selected DEPs in the enriched pathways for finding out the synergistic or antagonistic effect when in co-exposure with MPs. In a different comparison, like “SMZ+MP vs. SMZ group”, the term pathways of endocytosis, protein processing in the endoplasmic reticulum, and regulation of the actin cytoskeleton were enriched and selected. The selected target DEPs and their revealed descriptions are based on log2(fold change) and *p*-value. Green and red accession numbers stand for the down- and up-regulated DEPs.

**Figure 6 ijms-26-08441-f006:**
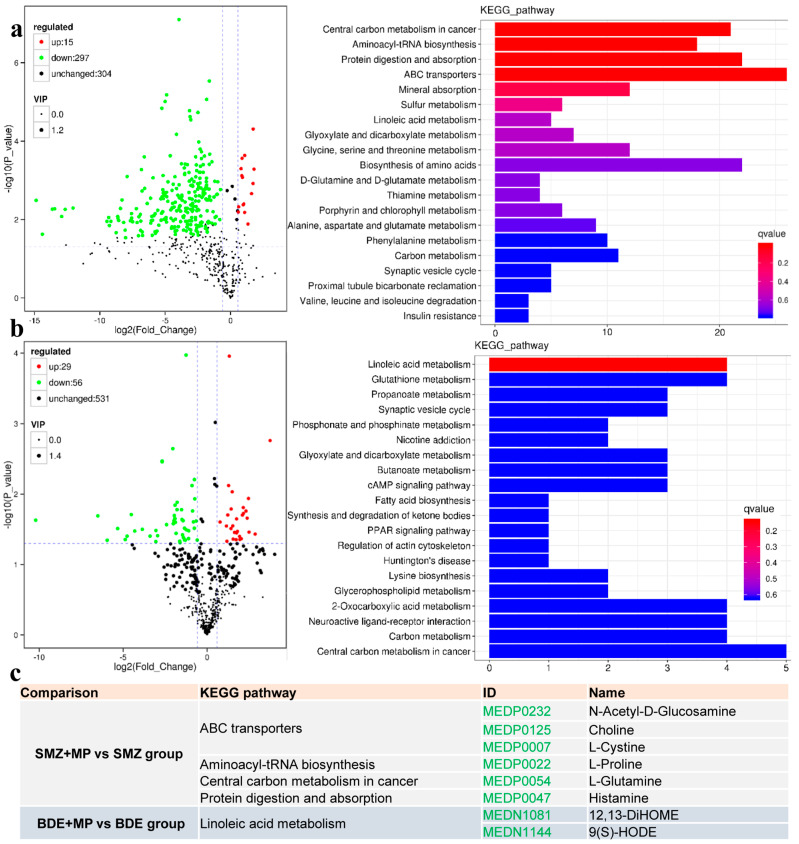
The metabolome results caused by MP, SMZ, and BDE_153_ co-exposure at 8 d (*n* = 6). (**a**) The volcano plot (**left**) and its related pathways (**right**) for the DEMs in the comparison (group SMZ+MP vs. SMZ) at 8 d, while (**b**) shows the comparison of group BDE+MP vs. BDE at 8 d. (**c**) The selected DEMs in the relative significant enriched pathway. In a different comparison, like “SMZ+MP vs. SMZ group”, the term KEEG pathways, IDs of DEPs, and its annotation names are revealed. The pathway of linoleic acid metabolism of the comparison “BDE+MP vs. BDE group”, 12,13-DiHOME, 12,13-dihydroxy-9Z-octadecenoic acid; 9(S)-HODE, 9(R,S)-hydroxy-10(E),12(Z)-octadecadienoic acid. Green accession numbers stand for the down-regulated DEMs.

**Figure 7 ijms-26-08441-f007:**
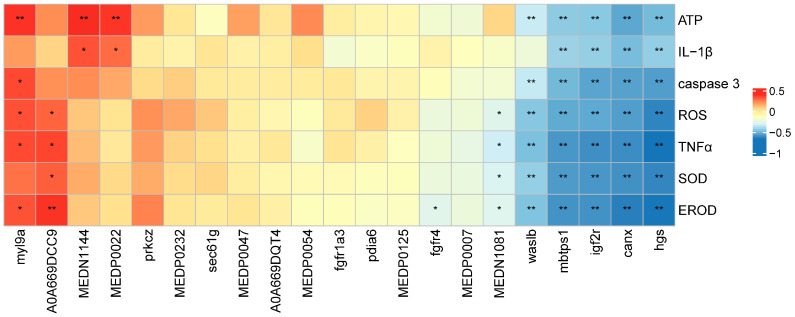
The correlation analysis between transcriptomics, proteomics, metabolomes, and enzymatic activities at 8 d. X and Y axes show genotoxicity, protein, metabolites, and different enzymatic activities. “*” and “**” show significant and extremely significant levels. Red and blue show positive and negative correlations among the selected parameters.

**Figure 8 ijms-26-08441-f008:**
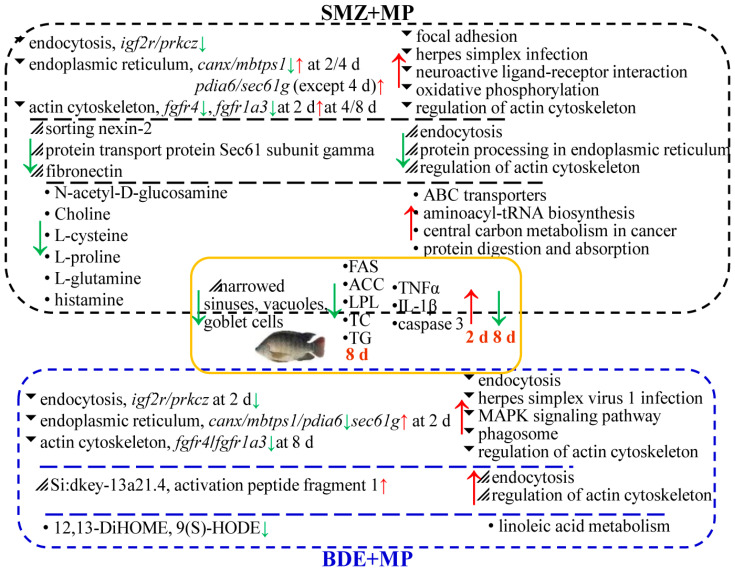
The experimental design and overall conclusion of this study. The biomarkers (left) and enriched KEGG pathways (right) in transcriptome, proteome, and fecal metabolome are ordered in the black dashed box for the comparison of “MP+SMZ vs. SMZ”. The blue dashed box reveals the biomarkers and pathways in the comparison of “MP+BDE vs. BDE”. The shared histological changes, enzymatic activities between the comparison of “MP+SMZ vs. SMZ” and “MP+BDE vs. BDE” are shown in the yellow box. “↑” and “↓” stand for the up- and down-regulated ones.

## Data Availability

The original contributions presented in this study are included in the article/Appendix A. Additional data supporting the findings of this study are available from the corresponding author upon reasonable request.

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
