# Peer review of "Multi-Omics Platforms Reveal Synergistic Intestinal Toxicity in Tilapia from Acute Co-Exposure to Polystyrene Microplastics, Sulfamethoxazole, and BDE153"

_ijms, 2025, doi:10.3390/ijms26178441_

Round 1
Reviewer 1 Report
Comments and Suggestions for Authors
Review of the manuscript titled “Integrated Multi-Omics Reveal Synergistic Intestinal Toxicity in Tilapia from Acute Co-exposure to Polystyrene Microplastics, Sulfamethoxazole, and BDE153”
The manuscript employs multiple omics technologies, including transcriptomics, proteomics, and metabolomics. However, it lacks clear evidence supporting a truly "integrated" multi-omics approach. Rather than integrating these platforms, the study presents results from each omics layer separately. Substantial improvements are needed before this work meets the publication standards of IJMS.
Weaknesses and Suggestions
- Abstract:
The abstract should be rewritten to be more scientifically structured, following a conventional format: Background / Objective / Methods / Results / Conclusions. It should clearly convey the study's key message in a concise and accessible manner. - Integration of Omics:
The manuscript does not explain how the different omics datasets were integrated. Please clarify the specific methods or bioinformatic strategies used to combine or cross-validate findings across transcriptomics, proteomics, metabolomics, and enzymatic assays. - Validation of Omics Findings:
The authors need to discuss how the omics findings were validated. Were any results confirmed using independent experimental approaches, such as qPCR, western blotting, or targeted assays? - Discrepancies Between Omics Layers:
How were potential discrepancies or conflicting results between transcriptomics, proteomics, metabolomics, and enzyme assays handled or interpreted? - Methodological Details:
The methods sections for transcriptomics, proteomics, and metabolomics lack sufficient detail. Please describe sample preparation, data acquisition, data preprocessing, normalization, and statistical analyses clearly for each modality. - Clarity of Results and Figures:
The results and associated figures are difficult to interpret. Currently, the figures appear to be pasted without adequate explanation, leaving the reader to infer the significance. Each figure should be described in sufficient detail in the main text to guide interpretation.
Supplementary Information - Quality control metrics include data completeness (before and after normalization), reproducibility, and technical variability.
- Full lists of differentially expressed in genes, proteins (DEPs), and metabolites, along with pathway enrichment results and functional annotations.
For example, please go through the whole manuscript
- The phrase "Proteomic DIA" should follow standard scientific terminology unless a novel technique is being described. If so, please define the method and its novelty.
- Line 23: The term "Si:dkey-13a21.4" is unclear. Is this a gene or protein identifier? Please clarify its biological relevance and provide context.
Author Response
Review#1
Review of the manuscript titled “Integrated Multi-Omics Reveal Synergistic Intestinal Toxicity in Tilapia from Acute Co-exposure to Polystyrene Microplastics, Sulfamethoxazole, and BDE153”
The manuscript employs multiple omics technologies, including transcriptomics, proteomics, and metabolomics. However, it lacks clear evidence supporting a truly "integrated" multi-omics approach. Rather than integrating these platforms, the study presents results from each omics layer separately. Substantial improvements are needed before this work meets the publication standards of IJMS.
Weaknesses and Suggestions
Abstract:
The abstract should be rewritten to be more scientifically structured, following a conventional format: Background / Objective / Methods / Results / Conclusions. It should clearly convey the study's key message in a concise and accessible manner.
Response: Thank you for the suggestions. The Abstract has been revised as followes, “Polystyrene microplastic (MPs) and its co-existing contaminants may exert different toxic effects on its surrounded aquatic organisms. In order to detect the intestinal harmful responses, tilapia were subjected to exposure with 75 nm MPs, 100 ng·L⁻¹ sulfamethoxazole (SMZ), 5 ng·L⁻¹ BDE153, and combinations thereof over periods of 2, 4, and 8 days. Enzymatic assays, transcriptomics, proteomics, and metabolomics were employed to evaluate intestinal histopathological effects. Results showed that significant reductions were observed in ATP, ROS, SOD, EROD, lipid metabolism-related enzymes, pro-inflammatory cytokines (TNFα, IL-1β), and apoptosis marker caspase 3 across all groups at day 8. Histological evaluation revealed diminished goblet cell density, with distinct vacuole formation in the BDE153+MPs group. KEGG pathway analysis highlighted disruptions in endocytosis, MAPK signaling, phagosome formation, and actin cytoskeleton regulation. Proteomic findings indicated notable enrichment in endocytosis (decreased sorting nexin-2; increased Si:dkey-13a21.4), MAPK/PPAR signaling, protein processing in the endoplasmic reticulum (Sec61 subunit gamma), and cytoskeletal modulation (reduced fibronectin; elevated activation peptide fragment 1), with or without SMZ and BDE153. Metabolomic profiling showed significant alterations in ABC transporters, aminoacyl-tRNA biosynthesis, protein digestion and absorption, and linoleic acid metabolism. Collectively in conclusion, these findings suggest that BDE153 and MPs synergistically exacerbate intestinal damage and gene/protein expression over time, while SMZ appears to exert an antagonistic, mitigating effect.”.
“Polystyrene microplastic (MPs) and its co-existing contaminants may exert different toxic effects on its surrounded aquatic organisms. In order to detect the intestinal harmful responses, T…Results showed that s…in conclusion” added in line 13-15, 18, 30.
Integration of Omics:
The manuscript does not explain how the different omics datasets were integrated. Please clarify the specific methods or bioinformatic strategies used to combine or cross-validate findings across transcriptomics, proteomics, metabolomics, and enzymatic assays.
Response: Thank you for the suggestions. The words “integrated” have been deleted in line 1, 97, 215, 382, 405, 519. Because the result for different omics datasets only showed in the correlation analysis, and especially with enzymatic activities in line 390. Its more method details about multiple omics have been added in line 159-163, 166-175, 178-185, 188-196, 200-208. qPCR verification for transcriptomics revealed in line 175 (also added in line 305), the revised version do not add something specific for other verification.
Validation of Omics Findings:
The authors need to discuss how the omics findings were validated. Were any results confirmed using independent experimental approaches, such as qPCR, western blotting, or targeted assays?
Response: Thank you for the suggestions. qPCR verification presented in line 175 and 305 with result in Fig.4 in line 312, with primers in line 556. Actually for the verification for the selected DEPs in Fig. 5 (line 337), the problem is some of them were difficult for annotation except for someones. The authors got the accesion IDs for such those DEPs, like sorting nexin-2 (XP_005451511.1 for isoform X1, and XP_003444550.1 for isoform X2), Sec61 subunit gamma (XP_013125823.1), fibronectin (XP_005450465.1), Si:dkey-13a21.4 (XP_003442681.2), without activation peptide fragment 1 (not found). The validation for metabolomics may need feed trial or with specific markings. The next step is to synthesize antibodies for validation experiments and report the corresponding results to the reviewers.
Discrepancies Between Omics Layers:
How were potential discrepancies or conflicting results between transcriptomics, proteomics, metabolomics, and enzyme assays handled or interpreted?
Response: Thank you for this valuable comment. We acknowledge the importance of clarifying how potential discrepancies between the different omics layers were addressed. In our study, no single gene, protein, and metabolite were found to be simultaneously and directly associated across all omics layers. However, convergence was observed at the pathway level. Specifically, in the PPAR signaling pathway, metabolites such as 12,13-DiHOME and L-tyrosine were associated with pathway activation, while IL-1β, 9(S)-HODE, and pparg were linked to lipid regulation via inflammatory responses. These observations suggest complementary roles rather than direct one-to-one correspondence between molecules across omics. Additional details are provided in lines 463–477 of the manuscript. Moreover, differentially expressed genes involved in lipid metabolism pathways have been validated and are presented in Figure 4 to support these integrative findings.
Methodological Details:
The methods sections for transcriptomics, proteomics, and metabolomics lack sufficient detail. Please describe sample preparation, data acquisition, data preprocessing, normalization, and statistical analyses clearly for each modality.
Response: Thank you for the suggestions. The more method details have been added in line 159-163, 166-175, 178-185, 188-196, 200-208.
Clarity of Results and Figures:
The results and associated figures are difficult to interpret. Currently, the figures appear to be pasted without adequate explanation, leaving the reader to infer the significance. Each figure should be described in sufficient detail in the main text to guide interpretation.
Response: Thank you for the suggestions. The more details on figures have been added in line 253-254, 257-260, 275-284, 289-291, 339-345, 374-378, 393-394, 417-422, 558-559.
Supplementary Information
Quality control metrics include data completeness (before and after normalization), reproducibility, and technical variability. Full lists of differentially expressed in genes, proteins (DEPs), and metabolites, along with pathway enrichment results and functional annotations.
Response: Thank you for the suggestions. The details for quality control metrics and those annotations have been added in line 166-175, etc.; line 253-254, 257-260, 275-284, 289-291, 339-345, 374-378, 393-394, 417-422, 558-559. The full list of DEGs, DEPs and DEMs revealed in the supplementary information excel.
Comments on the Quality of English Language. For example, please go through the whole manuscript. The phrase "Proteomic DIA" should follow standard scientific terminology unless a novel technique is being described. If so, please define the method and its novelty.
Response: Thank you for the suggestions. The DIA has been deleted in line 24, 165, 333. The authors have polished the revised ms. and revised in some places, like line 30, 96, etc.
Line 23: The term "Si:dkey-13a21.4" is unclear. Is this a gene or protein identifier? Please clarify its biological relevance and provide context.
Response: Thank you for the suggestions. "Si:dkey-13a21.4" was an annotation for protein, and this protein (https://www.ncbi.nlm.nih.gov/gene/494576/, proten name is ras-related protein rab7 or its domain-containing protein) has been published in two papers (Strausberg et al., 2002; Gaudet et al., 2011). The genes for its analogues have been published in zebrafish (si:dkey-25li10, Nie et al., 2021) and Atlantic Killifish (si:dkey-21c1.4, Albers et al., 2024).
References:
- Strausberg RL, Feingold EA, Grouse LH, et al., Mammalian Gene Collection Program Team. Generation and initial analysis of more than 15,000 full-length human and mouse cDNA sequences. Proc Natl Acad Sci U S A. 2002,99(26):16899-903. doi: 10.1073/pnas.242603899.
- Gaudet P, Livstone MS, Lewis SE, Thomas PD. Phylogenetic-based propagation of functional annotations within the Gene Ontology consortium. Brief Bioinform. 2011,12(5):449-62. doi: 10.1093/bib/bbr042.
- Nie CH, Zhang NA, Chen YL, Chen ZX, Wang GY, Li Q, Gao ZX. A comparative genomic database of skeletogenesis genes: from fish to mammals. Comp Biochem Physiol Part D Genomics Proteomics. 2021,38:100796. doi: 10.1016/j.cbd.2021.100796.
- Albers JL, Ivan LN, Clark BW, Nacci DE, Klingler RH, Thrash A, Steibel JP, Vinas NG, Carvan MJ, Murphy CA. Impacts on Atlantic Killifish from Neurotoxicants: Genes, Behavior, and Population-Relevant Outcomes. Environ Sci Technol. 2024,58(39):17235-17246. doi: 10.1021/acs.est.4c04207.
Reviewer 2 Report
Comments and Suggestions for Authors
The manuscript titled “Integrated Multi-Omics Reveal Synergistic Intestinal Toxicity in Tilapia from Acute Co-exposure to Polystyrene Microplastics, Sulfamethoxazole, and BDE153 (ijms-3704231)” by Zheng et al. found that BDE153 and MPs synergistically exacerbate intestinal damage and gene/protein expression over time, while SMZ appears to exert an antagonistic, mitigating effect through integrated multi-omics. The research is significant and the data presentation is largely satisfactory. Therefore, it is recommended that this research be published in the IJMS journal. However, there are several points that require clarification and revision.
(1) Introduction section: it provides a rather simplistic description of the research background, and it needs to be rewritten.
(2) The horizontal axis in Figure 1 requires a detailed explanation of its specific meaning. Such as, CK-2, SMZ-2…… means animals exposure for 2 days by PBS (control), SMZ……, respectively. And some of horizontal axis are missing in Figure 1C.
(3) Figure 2 is not clear. Please provide a clearer image with higher resolution.
(4) Line 216, the sentence of “8-day exposure named as 8AI, 8BI, 8CI, 8DI, 8EI, 8FI for 8 days, respectively” in the legend of Figure 3 are disordered. It was marked AI8, ect in the Figure 3.
(5) Figure 4, Please use the expression level of 3 samples from each group to create the heat map, rather than using only the average values.
Comments on the Quality of English LanguageNone.
Author Response
Review#2
The manuscript titled “Integrated Multi-Omics Reveal Synergistic Intestinal Toxicity in Tilapia from Acute Co-exposure to Polystyrene Microplastics, Sulfamethoxazole, and BDE153 (ijms-3704231)” by Zheng et al. found that BDE153 and MPs synergistically exacerbate intestinal damage and gene/protein expression over time, while SMZ appears to exert an antagonistic, mitigating effect through integrated multi-omics. The research is significant and the data presentation is largely satisfactory. Therefore, it is recommended that this research be published in the IJMS journal. However, there are several points that require clarification and revision.
- Introduction section: it provides a rather simplistic description of the research background, and it needs to be rewritten.
Response: Thank you for the suggestions. The Introduction has been revised as followes, “Polystyrene is one of the most extensively researched polymer materials, and its microplastic form (MPs) is recognized as an effective vector for various environmental pollutants, including antibiotics. These particles are widely dispersed in aquatic environments, with studies reporting average concentrations of 29 ng·L⁻¹ in groundwater near drinking-water sources [1]. Antibiotics, in comparison, have been frequently detected in surface waters and sediments at concentrations ranging from 1.12 to 377 ng·L⁻¹ and 7.95 to 145 ng·g⁻¹, respectively [2]. MPs and antibiotics are widely distributed in aquatic environments, and they can have deleterious effects on a wide range of aquatic species. Given their ubiquitous presence and ecological hazards, the co-occurrence of MPs and antibiotics has become a major environmental concern [1,3]. Several fish experiments as well as studies have shown that organismal damage such as intestinal damage and altered metabolic profles in fsh are associated with the bioaccumulation of MPs and related toxins. The phenomena of their coexistence occurs in a lot of study sites based on the collected published data [2]. Sulfamethoxazole (SMZ), a commonly detected antibiotic in aquatic sediments, has been associated with growth inhibition, reproductive disturbances, and physiological abnormalities in aquatic fauna [4]. SMZ has been detected ranging from ng·L⁻¹ to μg⋅L⁻¹, and especially in pond with a higher concentration as 273.20 ng·L⁻¹ in Taihu Lake [5]. Notably, SMZ in combination with MPs demonstrates heightened toxicity compared to individual exposures, as observed in medaka and zebrafish [5].
MPs are carriers of persistent organic pollutants (POPs), and capable of absorbing POPs. Several recent studies have assessed the combined effects of MPs and POPs, including antibiotics [1,5]. MPs can accumulate within the gastrointestinal tracts of aquatic animals and may translocate to systemic circulation [6]. POPs pose significant ecological risks due to their chemical stability, tissue-wide distribution, and potential for biomagnification through trophic levels—phenomena reported in the U.S., Canada, and China [7]. MPs also exhibit strong sorptive capacity toward organic pollutants such as polybrominated diphenyl ethers (PBDEs), thereby serving as carriers that can potentiate toxicological outcomes [8]. PBDEs are a class of organic pollutants characterized by low water solubility, high persistence, and widespread distribution [9], with a concentration of 44.0 ng⋅g−1 lipid in aquatic animals of China. Average concentrations of MP-affliated ∑8PBDE were 412 ng⋅g−1 in Pearl River Delta, South China, while ∑7PBDEs were from <LOD to 0.78 ng⋅g−1 in the sediment of Yangtze River [28]. Degradation of MPs may facilitate PBDE release into aquatic systems, where they have been found coexisting with crustaceans and beach sediments [9-10].
Environmental data indicate MP concentrations in sediment range from 44.42 to 417.56 items·kg⁻¹ [11], while MP-associated PBDEs reach up to 412 ng·g⁻¹ in the Pearl River Delta [12]. BDE153 (2,2′,4,4′,5,5′-hexabromodiphenyl ether), a major PBDE congener, has been detected in both human tissues via fish consumption and in aquatic species such as carp [13-14]. Tissue distribution studies show the highest BDE153 concentrations in bile, followed by brain, liver, gills, and muscle [15]. Smaller MPs (~27 μm) are particularly prone to intestinal accumulation [5]. Additionally, microbial biofilms on MP surfaces may enhance pollutant adsorption [16], and studies have identified strong correlations between MPs and total PBDE content. Our previous study showed 10 mg⋅mL−1 75 nm MPs resulted in hepatic damage possibly through PPAR signaling, endoplasmic reticulum pathway at 7–14 days [22]. When co-exposure with 5 ng⋅L−1 BDE153, the enzymatic activities of pro-inflammatory and apoptosis signifcantly increased, vacuoles appeared, pathways of endocytosis, regulation of actin cytoskeleton were signifcantly enriched [28].
Exposure to MPs and PBDEs can trigger oxidative stress, inflammation, apoptosis, and metabolic disruptions in aquatic organisms, as demonstrated in multi-omics studies [17-18]. BDE153 exposure has also been linked to neurotoxicity [19], and our previous study showed MPs and BDE153 co-exposure exerted a synergistic toxicity when compared to single exposure [28]. Notably, larger MPs tend to cause damage via indirect inflammatory responses, while nanosized MPs can directly enter cells and induce more severe effects [20]. Despite growing interest, the effects of different exposure durations remain unclear. Extended exposure may affect metabolism, immune responses, and gut microbiota, but these mechanisms are not fully understood [18,21]. Tilapia, a fsh species recommended by the FAO, is one of the most widely farmed species globally, with production exceeding 7000,000 tons, predominantly from China (1816,828 tons in 2023) and exported to worldwide. Therefore, the present study aimed to: (1) evaluate the acute intestinal toxicity of MPs, SMZ, and BDE153, both individually alone and in combination; and (2) elucidate the molecular mechanisms involved using an multi-omics approach including transcriptomics, proteomics, and metabolomics.”.
(2) The horizontal axis in Figure 1 requires a detailed explanation of its specific meaning. Such as, CK-2, SMZ-2…… means animals exposure for 2 days by PBS (control), SMZ……, respectively. And some of horizontal axis are missing in Figure 1C.
Response: Thank you for the suggestions. The annotation for those have been added in line 253-254, 257-260, and the missing horizontal axis has been added in line 250.
(3) Figure 2 is not clear. Please provide a clearer image with higher resolution.
Response: Thank you for the suggestions. Fig.2 has been replaced with higher resolution in line 266.
(4) Line 216, the sentence of “8-day exposure named as 8AI, 8BI, 8CI, 8DI, 8EI, 8FI for 8 days, respectively” in the legend of Figure 3 are disordered. It was marked AI8, ect in the Figure 3.
Response: Thank you for the suggestions. The name has been revised in line 127, 287-288.
(5) Figure 4, Please use the expression level of 3 samples from each group to create the heat map, rather than using only the average values.
Response: Thank you for the suggestions. The authors used figure based on the average values of each groups (i.e., SMZ stands for three sample for SMZ-2, SMZ-4, SMZ-8 in triplicate), in order to show the expression profile for our selected biomarkers in different groups, and further to reveal how the synergistic or antagonistic effects when co-exposure with SMZ and BDE, from which, the potential readers can be easily see the up-regulation or down-regulation based on the fold changes compared to the controls. For its reason, the authors suggested use the current one, and the revised one presened and located in the supplementary information Fig. A1 in line 557.

Round 2
Reviewer 1 Report
Comments and Suggestions for Authors
While the study presents interesting findings, the validation assays for protein expression and post-translational modifications should be conducted more rigorously to ensure data reliability. Given its current scope and depth, the study may be more appropriate for a different MDPI journal.
Author Response
#Reviewer1
While the study presents interesting findings, the validation assays for protein expression and post-translational modifications should be conducted more rigorously to ensure data reliability. Given its current scope and depth, the study may be more appropriate for a different MDPI journal.
Response: Thank you for the suggestions. Within 5 days’s revision peroid, the current study used the protein (Acyl-coenzyme A oxidase 1, which published in our previous paper, Li et al., 2023) and the metablity (L-Threonine, using standard solution) for validations, and the validation assays for protein expression and post-translational modifications supported our data, suggestions. The results showed as followes:
Fig. A2. Verifications for proteomics and metabolome among different groups. The name for proteomics and metabolome were Acyl-coenzyme A oxidase 1 (the value of the vertical axis equal to value from data/100000) and L-Threonine (the value of the vertical axis equal to value from data /10000000). The group name is the same to Fig.4. CK2, CK4, CK8 stands for the controls at 2, 4, 8 days respectively in triplicate.
References:
Li Q, Zheng Y, Sun Y, Xu G. Resveratrol attenuated fatty acid synthesis through MAPK-PPAR pathway in red tilapia. 2023. Comp Biochem Physiol C, 268:109598. 10.1016/j.cbpc.2023.109598.
The English could be improved to more clearly express the research.
Response: Thank you. The revised version has been polished by the English native speaker Ampere Yona, which revealed in blue.
